# Anti-Inflammatory Fibronectin-AgNP for Regulation of Biological Performance and Endothelial Differentiation Ability of Mesenchymal Stem Cells

**DOI:** 10.3390/ijms22179262

**Published:** 2021-08-26

**Authors:** Huey-Shan Hung, Kai-Bo Chang, Cheng-Ming Tang, Tian-Ren Ku, Mei-Lang Kung, Alex Yang-Hao Yu, Chiung-Chyi Shen, Yi-Chin Yang, Hsien-Hsu Hsieh, Shan-hui Hsu

**Affiliations:** 1Graduate Institute of Biomedical Science, China Medical University, Taichung 40402, Taiwan; kbwork2021@gmail.com (K.-B.C.); kutianren_0303@hotmail.com (T.-R.K.); 2Translational Medicine Research, China Medical University Hospital, Taichung 40402, Taiwan; 3College of Oral Medicine, Chung Shan Medical University, Taichung 55015, Taiwan; ranger@csmu.edu.tw; 4Department of Medical Education and Research, Kaohsiung Veterans General Hospital, Kaohsiung 43302, Taiwan; kungmeilang@gmail.com; 5Ministry of Health & Welfare, Changhua Hospital, Changhua 51341, Taiwan; yuchest71@gmail.com; 6Neurological Institute Head of Department of Neurosurgery, Taichung Veterans General Hospital, Taichung 407204, Taiwan; shengeorge@yahoo.com (C.-C.S.); jean1007@gmail.com (Y.-C.Y.); 7Department of Physical Therapy, Hung Kuang University, Taichung 433304, Taiwan; 8Basic Medical Education Center, Central Taiwan University of Science and Technology, Taichung 406053, Taiwan; 9Blood Bank, Taichung Veterans General Hospital, Taichung 407204, Taiwan; hhhsu@vghtc.gov.tw; 10Institute of Polymer Science and Engineering, National Taiwan University, Taipei 10617, Taiwan

**Keywords:** mesenchymal stem cells, fibronectin, silver nanoparticles, endothelial differentiation, vascular tissue engineering

## Abstract

The engineering of vascular regeneration still involves barriers that need to be conquered. In the current study, a novel nanocomposite comprising of fibronectin (denoted as FN) and a small amount of silver nanoparticles (AgNP, ~15.1, ~30.2 or ~75.5 ppm) was developed and its biological function and biocompatibility in Wharton’s jelly-derived mesenchymal stem cells (MSCs) and rat models was investigated. The surface morphology as well as chemical composition for pure FN and the FN-AgNP nanocomposites incorporating various amounts of AgNP were firstly characterized by atomic force microscopy (AFM), UV-Visible spectroscopy (UV-Vis), and Fourier-transform infrared spectroscopy (FTIR). Among the nanocomposites, FN-AgNP with 30.2 ppm silver nanoparticles demonstrated the best biocompatibility as assessed through intracellular ROS production, proliferation of MSCs, and monocytes activation. The expression levels of pro-inflammatory cytokines, TNF-*α*, IL-1β, and IL-6, were also examined. FN-AgNP 30.2 ppm significantly inhibited pro-inflammatory cytokine expression compared to other materials, indicating superior performance of anti-immune response. Mechanistically, FN-AgNP 30.2 ppm significantly induced greater expression of vascular endothelial growth factor (VEGF) and stromal-cell derived factor-1 alpha (SDF-1α) and promoted the migration of MSCs through matrix metalloproteinase (MMP) signaling pathway. Besides, in vitro and in vivo studies indicated that FN-AgNP 30.2 ppm stimulated greater protein expressions of CD31 and *von Willebrand Factor (vWF) as well as facilitated better endothelialization capacity than other materials. Furthermore, the* histological tissue examination revealed the lowest capsule formation and collagen deposition in rat subcutaneous implantation of FN-AgNP 30.2 ppm. In conclusion, FN-AgNP nanocomposites may facilitate the migration and proliferation of MSCs, induce endothelial cell differentiation, and attenuate immune response. These finding also suggests that FN-AgNP may be a potential anti-inflammatory surface modification strategy for vascular biomaterials.

## 1. Introduction

Various biomolecules used for the surface modification of biomaterials have been certified to ameliorate biocompatibility, cell proliferation, and to stimulate differentiation [1,2]. For patients with severe atherosclerotic cardiovascular diseases, replacement of blood vessels by the saphenous vein and mammary artery is the primary clinical approach [3]. Synthetic vascular grafts often fail because of at the implantation site and poor endothelialization capacity [4]. In order to conquer these barriers, functional endothelial cell (EC) seeding on vascular grafts was created [5]. Since the source of ECs is limited, appropriate biological performance and superior biocompatibility of synthetic grafts as well as high endothelialization capacity to reduce immune response after implantation is crucial to ensure the success of a long-term implant. In recent years, various surface modification methods were applied physically, chemically, and biologically to improve hemocompatibility [6,7]. Fibronectin (FN) is an abundant glycoprotein in the extracellular matrix (ECM) [8]. Previous research has indicated that fibronectin is distributed in connective tissue and contributes to angiogenesis and vascular remodeling [9,10]. Other literature also elucidates that the interaction between cells, growth factors, and FN can modulate appropriate microenvironment in tissue regeneration [11].

Immobilization of fibronectin on a salinized titanium surface was proved to facilitate fibroblast attachment [12] and enhance MG63 osteoblast-like cells’ differentiation capacity [13]. In addition, FN plays a critical role in constructing a temporary matrix for cell differentiation after the inflammatory phase [8]. This evidence suggests that FN may be employed for cardiovascular regeneration engineering.

Nanomaterials have unique properties [14] and, as biomaterials, they must be highly biocompatible to prevent any damage in organisms [15]. Nanocomposites are defined as filled polymers with nanofillers of a particle size smaller than 100 nm [16]. The combination of polymers and nanoparticles (NP) demonstrated diversified advantages on electrical or mechanical properties [17]. NPs such as gold nanoparticles (AuNP) and silver nanoparticles (AgNP) were widely applicable to biomedical treatments and tissue engineering [18]. AgNP are commonly used in wound repair management due to their superior anti-inflammatory properties and antimicrobial capacity that can enhance tissue healing [19]. A recent research indicated that a chitosan-silk fibroin decorating with AgNP-absorbed exosomes facilitated wound healing through angiogenesis, appropriate collagen deposition, and nerve regeneration in a rat model [20]. In addition, polyurethane (PU) composites containing different amounts of AgNP (~15.1, 30.2, and 75.5 ppm) revealed deflect surface features and, with 30.2 ppm of AgNP, demonstrated the superior biocompatibility [21], and promoted cell attachment, migration, proliferation, and differentiation of ECs [22]. The performance of PU-Ag nanocomposites was confirmed by a porcine model [23]. Meanwhile, the antibacterial activities of AgNP depend on surface oxidation and optimal particle dispersion [24]. The antimicrobial property of AgNP makes them an efficient candidate of surface coating for medical devices [25].

Stem cell homing, migration, and differentiation ability play important roles in vascular tissue regeneration [26]. Mesenchymal stem cells (MSCs) are a potential therapeutic tool due to self-renewal capacity, easiness for isolation, and low immunoreactivity [27]. The process of stem cell migration is regulated by specialized signals [28] and complicated interactions between adhesion molecules, chemokines, cytokines, and extracellular matrix-degrading proteases [29]. Moreover, the activated ECs express α_5_β_1_ and α_5_β_3_ integrins that can interact with ECM proteins such as fibronectin to regulate cell migration during vascular regeneration [30]. Meanwhile, stromal cell-derived factor α (SDF-1α) is a prominent chemokine to promote cell migration [31]. The SDF-1α and C-X-C chemokine receptor type 4 (CXCR4) signaling pathway is crucial in modulating stem cells migration [32], which activates focal adhesion kinase (FAK) and Rho GTPase, subsequently affecting matrix metalloproteinases (MMPs) activity and cell migration [33]. Moreover, various growth factors secreted by MSCs such as basic fibroblast growth factor (bFGF) and vascular endothelial growth factor (VEGF) effectively stimulate angiogenesis in the wound area [34]. Previously, AuNPs combined with fibronectin (FN-Au) were found to induce VEGF and enhance cell migration via endothelial nitric oxide synthase (eNOS)/MMP signal pathway [35]. However, vascular grafts-induced restenosis and thrombosis frequently occur after implantation [36]. Thus, novel vascular biomaterials that can suppress foreign body response such as inflammation, thrombus, and neointimal hyperplasia causing in-stent restenosis are highly demanded [37].

FN combined with type I collagen, VEGF and SDF-1α was reputed to improve endothelialization [38]. A novel biostable polyurethane elastomer (TPCU) biofunctionalized with FN and decorin (DCN) did not induce major immune responses, and it could attract endothelial colony-forming cells (ECFCs), suggesting the potential of FN to be an appropriate natural biomolecule for creating nanocomposites [39]. Previously, FN-Au nanocomposites were found to reduce monocyte activation and enhance biocompatibility [35,40]. In the current study, AgNP combined with fibronectin (FN) because the polyurethane–Ag nanocomposites were observed to have better hemocompatibility than polyurethane–Au nanocomposites [21,41]. In vitro and in vivo evaluation showed FN-AgNP to be a novel antimicrobial and anti-inflammatory surface modification coating for vascular repair and regeneration engineering.

## 2. Results

### 2.1. Characterization of FN and FN-AgNP

The surface morphology of AgNP was observed by SEM analysis (Figure 1A). The nanoparticle size of AgNP is about 5 nm. Furthermore, TEM imaging also confirmed the diameter of AgNP was about 5 nm as shown in Figure 1B. Indeed, the nanotopography were observed by AFM assay as shown in Figure 1C. The value of root mean square (Rq) and average roughness (Ra) were semi-quantified and indicated as Figure 1D. The value of Rq and Ra were significantly higher at 30.2 ppm of FN-AgNP (Rq: ~1.16 nm, Ra: ~0.85 nm), compared to FN-AgNP 75.5 ppm (Rq: ~0.77 nm, Ra: ~0.52 nm), FN-AgNP 15.1 ppm (Rq: ~0.77 nm, Ra: ~0.60 nm) and pure FN (Rq: ~0.67 nm, Ra: ~0.50 nm). FN is an adhesion molecule that composes of three different repeating domains, and contains amino-terminal FN dimers which can be assembled into ECM of fibroblasts. The above results indicated that FN and AgNP may facilitate cellular interactions with ECM contributing to effect differentiation of stem cells. The UV-Vis absorption peak at 407 nm was attributed to the increasing concentration of AgNP from 15.1 ppm to 75.5 ppm (Figure 1E). According to Figure 1F, the position of the amide-I band maximum was at approximately 1636 cm^−1^ [42]. When FN was combined with AgNP, there was a shift in the peak position of amide I from 1623.98 cm^−1^ (pure FN) to 1622.21 cm^−1^ (FN-AgNP 15.1 ppm), 1621.61 cm^−1^ (FN-AgNP 30.2 ppm) and 1621.56 cm^−1^ (FN-AgNP 75.5 ppm). The above findings indicated that the amide I band may have strong interaction with AgNP. [43]. After the combination of AgNP into pure FN, significant changes occurred in the spectra near 1610~1700 cm^−1^ (peptide carbonyl stretching vibration, CO). According to Figure 1F, there was a shift in the peak position of amide I from 1623.98 cm^−1^ (pure FN) to 1622.21 cm^−1^ (FN-AgNP 15.1 ppm), 1621.61 cm^−1^ (FN-AgNP 30.2 ppm) and 1621.56 cm^−1^ (FN-AgNP 75.5 ppm). The above findings indicated that the amide I band may have strong interaction with AgNP.

### 2.2. Cytoskeletal Change and Migration Ability of HSF and MSC on FN-AgNP Nanocomposites

It has been reported that co-culture of fibroblasts and endothelial could significantly enhance the angiogenesis of endothelial cells. It was indicated that there are intimate communications between fibroblasts and endothelial cells for angiogenesis, and the paracrine effect may play a crucial role in these communications [44]. Therefore, the effects of different materials on biocompatibility and the biological activity of fibroblast (HSF) were also investigated in this study. The assessment of actin fiber staining via phalloidin-rhodamine conjugate is demonstrated in Appendix A and Figure 2A. In control group, the actin fibers of HSF and MSC were mostly near the cell body and exhibited in circumferential shape. While seeding cells with FN-AgNP 30.2 ppm, the actin fibers became more extended. The length of actin fiber was much longer in MSC culturing in FN-AgNP nanocomposites, especially in FN-AgNP 30.2 ppm (~1.46-fold) and following by FN-AgNP 15.1 ppm (~1.35-fold), pure FN (~1.23-fold) and FN-AgNP 75.5 ppm (~1.20-fold). The above condition could also be observed in HSF, but had a less extension comparing to MSCs: FN-AgNP 30.2 ppm (~1.39-fold), FN-AgNP 15.1 ppm (~1.28-fold), pure FN (~1.20-fold) and FN-AgNP 75.5 ppm (~1.15-fold) (Appendix A and Figure 2C).

As Appendix A and Figure 2B showed, the real-time observation of cell migration distance was observed in pre-migration (t = 0 h) and post-migration (t = 24 and 48 h). The semi-quantitative data (Figure 2D and Appendix A) in MSCs from 24 to 48 h on FN-AgNP 30.2 ppm (~1.24 to ~1.46-fold) was significantly greater than other materials [FN-AgNP 15.1 ppm (~1.18 to ~1.39-fold), FN-AgNP 75.5 ppm (~1.10 to ~1.22-fold) and pure FN (~1.07 to ~1.18-fold)]. The similar condition could be observed in HSF particularly in FN-AgNP 30.2 ppm (~1.28 to ~1.46-fold), following by FN-AgNP 15.1 ppm (~1.20 to ~1.35-fold), FN-AgNP 75.5 ppm (~1.15 to ~1.25-fold) and pure FN (~1.12 to ~1.22-fold). Based on the results, FN-AgNP 30.2 ppm could induce advanced cell adhesion and migration behaviors.

### 2.3. Cell Proliferation and ROS Production

The proliferation of HSF and MSCs culturing on various materials for 24 and 48 h are demonstrated in Appendix A and Figure 2E. The cell viability of HSF (OD_570nm_ = ~0.92) and MSC (OD_570nm_ = ~0.80) at 48 h was both the highest on FN-AgNP 30.2 ppm, following by FN-AgNP 15.1 ppm (HSF: OD_570nm_ = ~0.83, MSC: OD_570nm_ = ~0.70), FN-AgNP 75.5 ppm (HSF: OD_570nm_ = ~0.69, MSC: OD_570nm_ = ~0.69) and pure FN (HSF: OD_570nm_ = ~0.66, MSC: OD_570nm_ = ~0.68). ROS generation of HSF and MSC on various materials is depicted in Appendix A. At 48 h, the FN-AgNP 30.2 ppm induced the lowest amount of ROS generation in both HSF (~0.5-fold) and MSC (~0.42-fold), comparing to FN-AgNP 15.1 ppm (HSF: ~0.7-fold, MSC: ~0.65-fold), FN-AgNP 75.5 ppm (HSF: ~0.73-fold, MSC: ~0.7-fold) and pure FN (HSF: ~0.87-fold, MSC: ~0.8-fold).

### 2.4. Biocompatibility Assay

The monocyte activation was further evaluated by CD68 staining (the marker of macrophage) shown in Figure 3A. The quantification of CD68 expression resulting from fluorescent intensity was demonstrated in Figure 3B. While in FN-AgNP 30.2 ppm (~0.52-fold), the expression of CD68 was the lowest, following by FN-AgNP 15.1 ppm (~0.52-fold), FN-AgNP 75.5 ppm (~0.60-fold) and pure FN (~0.88-fold). And a similar trend of conversion ratio among different materials was shown in Figure 3C. Furthermore, the pro-inflammatory cytokines (TNF-*α*, IL-1β, IL-6) expression were also examined via ELISA assay (Figure 3D). The quantitative data indicated the lowest expression in FN-AgNP 30.2 ppm. These results suggested that FN-AgNP nanocomposites may stimulate lower inflammatory response.

### 2.5. Effect of FN-AgNP Nanocomposites on SDF-1α Expression and MMPs Activation in MSCs

The representative zymograms for MMP-2 (62 kDa) and MMP-9 (90 kDa) at various times (24 and 48 h) are displayed as Figure 4A. The semi-quantification of MMP-9 gene expression is represented in Figure 4B. Indeed, the expression of MMP-9 in MSCs at 48 h is the greatest in FN-AgNP 30.2 ppm (~1.29-fold), following by FN-AgNP 15.1 ppm (~1.22-fold), FN-AgNP 75.5 ppm (~1.20-fold) and pure FN (~1.11-fold). On the basis of previous study, the SDF-1α protein is investigated which plays a critical role in MSCs migration for promoting tissue repair. Semi-quantitative results (Figure 4C) showed that after 24 and 48 h of incubation, the SDF-1α expression in MSCs on FN-AgNP 30.2 ppm (~1.22-fold to ~1.28-fold) examined by ELISA assay was significantly greater than other groups [FN-AgNP 15.1 ppm (~1.19-fold to ~1.2-fold), FN-AgNP 75.5 ppm (~1.11-fold to ~1.12-fold) and pure FN (~1.1-fold to ~1.12-fold)]. The MMP-9 expression in HSF was also showed in Appendix A [FN-AgNP 30.2 ppm (~1.34-fold to ~1.44-fold), FN-AgNP 15.1 ppm (~1.26-fold to ~1.3-fold), FN-AgNP 75.5 ppm (~1.4-fold to ~1.28-fold) and pure FN (~1.2-fold to ~1.22-fold)].

### 2.6. Endothelialization of MSCs Induced by FN-AgNP Nanocomposites

Immunofluorescence staining analysis can be clearly observed in Figure 5A,B, to characterize the expression of EC marker (CD31 and vWF) after seeded MSCs on FN-AgNP for 7 days. Figure 5C,D demonstrate the semi-quantitative data of CD31 and vWF expression. The CD31 and vWF expression on FN-AgNP 30.2 ppm (CD31: ~1.52-fold, vWF: ~1.47-fold) are significantly greater than other groups [FN-AgNP 15.1 ppm (CD31: ~1.40-fold, vWF: ~1.34-fold), FN-AgNP 75.5 ppm (CD31: ~1.24-fold, vWF: ~1.29-fold) and pure FN (CD31: ~1.25-fold, vWF: ~1.22-fold)]. The above results strongly suggest that FN-AgNP nanocomposites may facilitate MSC differentiate into ECs. The differentiation capacity of MSCs induced by various materials was then evaluated. The phenotypes of endothelial cells were also observed at days three and five. After culturing MSCs with various materials, FN-Ag 30.2 ppm induced a slight expression level of endothelial markers at days three and five as compared to the control, with the immunostaining images shown in Appendix A. The semi-quantitative data showed that the mineral? differentiation in the FN-Ag 30.2 ppm group was the greatest at days three and five (*p* < 0.01) (Appendix A).

### 2.7. Biocompatibility and Endothelialization Ability in the Rat Subcutaneous Model

Indeed, chronic inflammation may cause fibrosis and then leads to poor tissue regeneration and affect biocompatibility of nano biomaterials. In brief, a successful implantation is commonly determined by the interaction of cells and molecules between the implanted body and host tissue. The fibrotic encapsulation caused by foreign body response from the different materials was observed via subcutaneous after one month of implantation in order to confirm the biocompatibility and biosafety in vivo (Figure 6A). Indeed, it was also calculated the intensity of tissue fibrosis effect using Masson’s trichrome staining, which revealed collagen deposition in response to the control treated group (glass) (Figure 6B). The semi-quantification analysis results are represented in Figure 6C,D. The value of capsule thickness was the lowest in FN-AgNP 30.2 ppm group (~0.3-fold), comparing to FN-AgNP 15.1 ppm (~0.4-fold), FN-AgNP 75.5 ppm (~0.45-fold) and pure FN (~0.92-fold). And the collagen deposition was also investigated and had a similar trend with capsule thickness [FN-AgNP 30.2 ppm (~0.34-fold), FN-AgNP 15.1 ppm (~0.4-fold), FN-AgNP 75.5 ppm (~0.5-fold) and pure FN (~0.78-fold)]. In addition, the endothelialization marker CD31, and VEGF were also investigated (Figure 7). As Figure 7A,B showed, the expression of CD31 was significantly higher in FN-AgNP 30.2 ppm (~1.46-fold) group than in other materials [FN-AgNP 15.1 ppm (~1.26-fold), FN-AgNP 75.5 ppm (~1.12-fold) and pure FN (~1.13-fold)]. And the expression of VEGF was measured at 24 and 48 h through ELISA assay (Figure 7C). The results showed the expression in FN-AgNP 30.2 ppm group (~1.23-fold to ~1.33-fold) was greater than other materials [FN-AgNP 15.1 ppm (~1.12-fold to ~1.23-fold), FN-AgNP 75.5 ppm (~1.12-fold to ~1.22-fold) and pure FN (~1.08-fold to ~1.11-fold)]. Furthermore, the macrophage polarization was examined and demonstrated as Figure 8A,B. The marker for M1 polarization, CD86, was significantly lower expressed in FN-AgNP 30.2 ppm group (~0.22-fold), then followed by FN-AgNP 75.5 ppm (~0.38-fold), FN-AgNP 15.1 ppm (~0.44-fold) and pure FN (~0.82-fold) groups (Figure 8C). Next, CD163 was selected as t M2 polarization marker, then found a reverse trend. Indeed, the tissues in the FN-AgNP 30.2 ppm group (~1.42-fold) was the greatest expression, comparing to FN-AgNP 15.1 ppm (~1.32-fold), FN-AgNP 75.5 ppm (~1.20-fold) and pure FN (~1.12-fold) (Figure 8D). The above results elucidated FN-AgNP 30.2 ppm could remarkably suppress inflammatory responses and enhance endothelialization ability. The biocompatibility capacity, biological performance, endothelial differentiation ability, and anti-inflammatory response was enhanced by FN-AgNP nanocomposites both in in vivo and the in vivo assay as illustrated in Figure 9.

## 3. Discussion

Natural polymeric grafts demonstrate no cell toxicity and low immune response in vascular tissue engineering [45]. Therefore, biocompatibility is the focal point while designing appropriate biomaterials for implantation in human body. To take the advantages of the better mechanical properties of the synthetic polymers, and for the purpose of improving biocompatibility, various surface modification methods such as chemical, physical, and biomolecule immobilization have been used. Among the methods, changing the material surface topography on a nanoscale may affect surface properties, such as protein, cellular, and tissue interactions, thereby improving the performance of the biomaterials. Introduction of roughness (Rq and Ra) of biomaterials in nano-scales can promote cell adhesion [46,47]. Surface modification by several proteins such as collagen, elastin, and fibronectin also mediate vascular cell interactions [46].

Adding an optimized concentration of AgNP in the natural polymers can generate surface morphology favorable for cell interaction [23]. The current research focused on creating surface functionalized coatings comprised FN and AgNP to improve cell behavior and biocompatibility. Based on AFM results, the value of surface roughness was the highest in FN-AgNP 30.2 ppm, confirming that the surface morphology of pure FN was remarkably changed by adding optimal concentration of AgNP. Besides, FTIR results demonstrated the interaction between AgNP and amide I region (peptide carbonyl stretching vibration, CO) of FN in the FN-AgNP nanocomposites. The above evidence demonstrated that the physical-chemical property of pure FN was altered after decorating with Ag nanoparticles.

FN-AgNP 30.2 ppm stimulated the attachment, proliferation, and migration of MSCs in vitro. In addition, FN-AgNP 30.2 ppm showed better biocompatibility including the reduction of monocyte activation and pro-inflammatory cytokines (TNF-*α*, IL-1β, IL-6) expression [48,49]. In other words, the addition of AgNP attenuated the inflammatory response of biomaterials. Anti-oxidation is related to reduced inflammation after implantation of nano-biomaterials [50]. A pervious study indicated that after implanting antioxidant-modified polymer into rat models, the acute inflammatory infiltrates were reduced [51]. Besides, FN-AgNP inhibited ROS generation in MSCs based on the results. The above findings supported that AgNP are potent to improve the biological performances of FN. The differentiation ability of stem cells can be stimulated by biomolecules or an appropriate ECM microenvironment. Apart from the support of scaffolds, suitable ECM can promote advanced biological responses such as cell viability, proliferation, and migration through activating cell signaling pathways [52]. MSCs are potential cell source in tissue repair engineering owing to secretion of various bioactive factors and superior differentiation capacity for therapeutic treatments [53]. The current study demonstrated that FN-AgNP 30.2 ppm could regulate cellular morphology and biological functions of MSCs. The results of the endothelialization marker expression, CD31, indicating that FN-AgNP 30.2 ppm had the ability to induce endothelial cells differentiation. The above findings of promoted EC phenotype and better biocompatibility for MSCs seeded on FN-AgNP suggested the efficiency of FN-AgNP nanocomposites to be a suitable biomaterial for vascular tissue regeneration treatments.

As mentioned above, the superior proliferation, migration, and differentiation ability of MSCs coupled with biomaterials are indispensable for wound repair and regeneration. The MMPs secretion is associated with promoting vascular remodeling and cell migration [54]. The expression of MMP-2/9 contributes to cell migration through activating cell surface molecules such as α5β3 integrin [55]. Various angiogenic cytokines such as VEGF and SDF-1α can modulate and improve the process of tissue regeneration [56]. The VEGF and SDF-1α protein expression in MSCs can facilitate vasculogenesis and angiogenesis. Our previous study demonstrated that polyurethane-AuNPs (PU-AuNPs) could regulate ECs migration via triggering α5β3 integrin/FAK and PI3K/Akt/eNOS signaling pathways [57]. In the current study, MMP activity, VEGF, and SDF-1α protein expression were investigated. The results showed that FN-AgNP 30.2 ppm could induce the highest expression of angiogenic cytokines and MMP proteinases. Owing to VEGF induces stem cells trafficking through the activation of SDF-1α/CXCR4 signaling pathway, the potential of FN-AgNP to recruit MSCs may be the advantage as vascular tissue repair biomaterials [29]. Based on the results of IHC and immunofluorescence staining, the implantation of MSCs cultured with FN-AgNP 30.2 ppm decreased M1 recruitment (CD86), but increased M2 infiltration in contrast. Besides, FN-AgNP 30.2 ppm also caused the lowest fibrous encapsulation and collagen deposition, indicating the better biocompatibility of FN-Ag 30.2 and the best tissue integrity after implantation. The above observation pointed out that surface modification of biomaterial may influence foreign body response in vivo.

Bacterial infection during intravenous injection is a barrier for clinical treatments. The short-term peripheral venous catheters (PVC) are one of the commonly used invasive medical devices in hospitals, but PVC often fail due to vascular or infectious complications such as gram-negative bacteremia or other bloodstream infections before the end of treatments. The above reasons can cause the interruption of treatments that is harmful to patients. Moreover, PVC replacement causes pain on patients and increases treatment costs [58]. A biodegradable water absorption sponge incorporating AgNP had antibacterial properties [59]. Likewise, FN-AgNP nanocomposites can become a novel of surface modification biomaterials for clinical treatments. After coating FN-AgNP on PVC, it may have potential to prevent bloodstream infection and reduce inflammatory response on patients. This kind of surface modified peripheral venous catheters may be developed as a long-term injection medical device to reduce failure rate and decrease treatment costs.

In conclusion, this research suggests that via making composites with fibronectin and AgNP, the behavior of MSCs on the biomaterial such as protein expression of SDF-1α and MMP enzymatic activities can be affected, leading to vascular remodeling. The nanocomposite coating improved blood compatibility, enhanced MSCs migration, proliferation, and EC phenotype as well as attenuated the immune response and ROS generation. With these benefits, the FN-AgNP nanocomposite may be a promising nanobiomaterial to provide a suitable surface coating and delivery technology for implanted cardiovascular devices.

## 4. Materials and Methods

### 4.1. Material Preparation of Fibronectin-Silver (FN-AgNP) Nanocomposite

The solution of human fibronectin was obtained through Millipore Corporation (USA) and the silver nanoparticles (AgNP) were purchased from GOLD NANOTECH, INC (Taiwan). The AgNP were dispersible into distilled water (~17.5 ppm). The size of silver nanoparticles could be mechanically controlled which was described in our previous study, and the diameter of the AgNP was approximately 5 nm [60]. The preparation of FN-AgNP composites was mixed with pure FN solution (1 mg/mL) and various concentration of AgNP (~15.1, ~30.2 and ~75.5 ppm). The appropriate amount of pure FN and FN-AgNP nanocomposites at 20 μg/cm^2^ were then coated on cell culture tissue dishes, culture plates, or round coverslip glasses (15 mm). The solution was adsorbed to the surface of each culture area for 30 min. Afterwards, the residual solution was removed to form FN and FN-AgNP thin films prior to proceeding further cell culture standard operating procedures for following experiments.

### 4.2. Material Characterization of FN and FN-AgNP Nanocomposites

Surface morphology of AgNP was observed using scanning electron microscope (SEM) (JEOL JEM-5200, Akishima, Tokyo, Japan). Surface topography were captured and the value of average roughness was further measured by Image J 5.0 software. TEM images were taken by transmission electron microscope (JEOL JEM-1010, Akishima, Tokyo, Japan) with the voltage setting up at 80 keV, then observed microstructure and particle size. For TEM imaging, 5 μL of nanoparticle solution was added onto copper grid then dried out at RT. Atomic force microscope (AFM) coupled with a scanner (100 μm, JEOL JSM-6700 F) was used to observe the surface morphology of FN and FN-AgNP dry coatings. Then the AFM measurements were processed through tapping mode with force constants of 21–78 N/m in air. The absorption spectrum of pure FN and FN-AgNP solutions were analyzed by UV-Vis spectrophotometer (Helios Zeta, USA). Fourier transform IR spectroscopy (IR Pretige-21, Shimadzu, Tokyo, Japan) was applied to analyze the chemical composition of each material. All samples were independently scanned for 8 times in the 400–4000 cm^−1^ spectral region with a 2 cm^−1^ resolution to obtain each spectrum.

### 4.3. Cell Proliferation Test

Human umbilical cord Wharton’s jelly-derived mesenchymal stem cells [61] were harvested cautiously for the following in vitro and in vivo assay. The MSCs were cultured in high glucose DMEM medium (Invitrogen) with 10% FBS, 1% (*v/v*) antibiotics P/S and 1% of sodium pyruvate. Besides, MSCs maintained correct phenotypes at passages 8–20 [33]. Human skin fibroblasts (HSF) were purchased from American Type Culture Collection (ATCC). They can be cultured to higher passage numbers without appreciable loss of growth rate or phenotype, thus yielding more cells for the experiments. HSF were maintained in Dulbecco’s Modified Eagle’s Medium (DMEM) (Gibco) supplemented with 1% (*v/v*) antibodies (10,000 U/mL penicillin G and 10 mg/mL streptomycin), 2 mM glutamine, and 10% fetal bovine serum (FBS). 200 μL of cells was seeded into the bottom of a 96-well plate with pure FN or FN-AgNP coatings at the density of 6 × 10^3^ cells/mL in each well. The cells seeded in blank well were represented as the control. The cells were further examined by MTT assay to evaluate cell viability after 24 and 48 h of incubation. The MTT [3-(4, 5-cimethylthiazol-2-yl)2, 5-diphenyltetrazolium bromide] solution (0.5 mg/mL) was added into each well for 3 h incubation (37 °C, 5% CO_2_). Afterwards, the crystals were dissolved in dimethyl-sulfoxide (DMSO) solution and the absorbance was measured at 570 nm using SpectraMax M2 microplate reader (Molecular Devices, San Jose,CA, USA).

### 4.4. Assessment of Reactive Oxygen Species (ROS) Generation

The ROS production was examined by 2, 7dichlorofluorescin diacetate (DCF-dA) oxidation-sensitive fluorescent dye (Sigma). The MSCs (2 × 10^5^/well) were firstly cultured in culture plates (6 well) with pure FN or FN-AgNP coatings then incubated for 24 and 48 h. Afterwards, the cells were washed two steps with PBS and added PBS containing DCF-dA (20 μM, 500 μL) for 1 h incubation. Hereafter, FACS Calibur flow cytometer (Becton Dickinson) was applied to investigate the MSCs at 530 nm after excitation at 480 nm for 4 h. The ROS production was demonstrated an increase trend based on fluorescence intensity. The positive cells were detected by the FCS software.

### 4.5. Monocyte Activation

Monocytes were obtained from whole blood of healthy volunteers through Percoll density gradient centrifugation (Sigma, St. Louis, MO, USA) and culturing by the standard procedure [35]. The cell concentration was adjusted to 1 × 10^5^ cells/mL. Cells were stored in RPMI medium [10% FBS, 1% (*v/v*) antibiotic (10,000 U/mL penicillin G and 10 mg/mL streptomycin)]. Twenty-four well tissue culture plates with pure FN and FN-AgNP nanocomposites coatings were firstly prepared, then 1 mL of cell suspension (1 × 10^5^/^mL^) was sequentially added into each well and incubated for 96 h to reach adherent (5% CO_2_). Next, the cells were collected by *trypsinization*. Based on the different cell morphology, an inverted microscope was further applied to calculate the numbers of macrophages and monocytes. The conversion ratio was then semi-quantified.

### 4.6. Cytoskeleton of MSC

In order to observe the cytoskeleton of MSCs, cells (5 × 10^3^/well) were cultured in 24 well plate containing pure FN and FN-AgNP on coverslip glass then incubated for 48 h. Cells culturing on blank coverslip glass was represented as the control group. Before the staining, MSCs were fixed by 4% paraformaldehyde (PFA) coupled with Phosphate Buffered Saline (PBS) for at least 15 min, then permeabilized with 0.5% (*v/v*) Triton-X 100 (Sigma) in PBS for another 10 min. To block non-specific interactions, 5% (*v/v*) Bovine Serum Albumin (Sigma) solution was used for the experiment. Ultimately, Rhodamine/phalloidin (6 μM, 30 min) and DAPI staining (1 μg/mL, 10 min) were performed to analyze cytoskeleton and nuclei in MSCs. The length of F-actin fiber in MSCs was measured as the ratio of the length of the existing active fiber to its potentially possible length restricted by cell margins. Length of F-actin fiber in MSCs were quantified using Image J (version 1.8.0_172) software by obtaining data from five different positions for each cell in the measurement (*n* = 3).

### 4.7. Assessment of MSCs Migration

A Oris Cell Migration Assay kit (Platypus Technologies, Madison, WI, USA) was applied to investigate cell migration. coating with pure FN and FN-AgNP were put into each well of the plates. Cell seeding stoppers (diameter 2 mm) was placed in the central area on the top of coverslip glasses, to prevent cells adhere to the center area of the well. The stopper in each well was seeded with the cell density of 8000 MSCs (100 μL/well) and incubated for 24 and 48 h to reach enrichment. Then the stoppers were removed from the culture plate, but remaining in several wells to represent as the pre-migration references. The seeded plates were incubated at the condition 37 °C/5% CO_2_ then observed at 0 h for pre-migration, at 24 and 48 h for post-migration. Next, 200 μL of Calcein AM (Sigma, USA) (2 μM) solution supplemented with serum-free culture medium was added to the plates, then images were taken through a fluorescence microscope (Zeiss Axio Imager A1, White Plains, NY, USA). The migration distance of MSCs in detection area was semi-quantified based on fluorescence intensity and Image J 5.0 software.

### 4.8. Immunofluorescence Analysis of vWF, CD31 and CD68 Expression

The MSCs (20,000 cells) were cultured in 24 well plate containing pure FN and FN-AgNP coatings on coverslip glass. The cultured cells were further fixed and permeabilized using PBS solution. The primary antibodies [anti- von Willebrand Factor (vWF), anti-CD31 (1:300 dilution) and anti-CD68 (1:150 dilution, Santa Cruz) antibodies] were used for overnight incubation. Then the cells need further washing steps. Afterwards, the secondary immunoglobulin, FITC-conjugate (1:300 dilution) was added for 1 h incubation. DAPI (1 mg/mL, Invitrogen) was added to locate cell nuclei then incubated for 10 min. Each cell suspension was added on microscope slides containing glycerol/PBS solution then cautiously sealed with synthetic mount after washes for further observation.

### 4.9. Enzyme-Linked Immunosorbent Assay

Several ELISA kits (R&D Systems, USA) were purchased to investigate the expression of SDF-1α, VEGF and proinflammatory cytokines (TNF-*α*, IL-1β, IL-6). The proteins were enriched in secretion by MSCs for the following measurements. The protein samples collected from MSCs were firstly detected with the primary antibody. Next, secondary antibody (HRP-conjugate) was used for signal amplification following with the manufacture’s instruction. Then the semi-quantitative results were analyzed through an ELISA reader. Quadruple experiments were proceeded to further obtain the data.

### 4.10. Gelatin Zymography Assay

The MSCs (2 × 10^5^ per well) were cultured in 6 well plate and incubated overnight to reach attachment. After 48 h, the culture medium was collected. The samples were separated by 10% SDS-PAGE with 2% gelatin, and the gel incubated with a pH 8.5 denaturing buffer (40 mM Tris-HCl, 0.2 M NaCl, 10 mM CaCl_2_, and 2.5% Triton X-100) for 30 min at RT. The gel was then slowly stirred at room temperature and equilibrated with a pH 8.5 development buffer (40 mM Tris- HCl, 0.2 M NaCl, 10 mM CaCl_2_, and 0.01% NaN_3_) for 16–18 h to become activated in 37 °C water bath. Afterwards, the gel was stained with 0.2% Coomassie Brilliant Blue R-250 (10% acetic acid, 50% methanol) and further washed with destaining buffer (10% acetic acid, 20% methanol). After Coomassie blue staining, the protease-digested gelatin area appeared as clear bands against blue-stained background. Hereafter, resulting gel was scanned in a densitometer and MMP gelatinase activity was semi-quantified through Image J 5.0.

### 4.11. Rat Subcutaneous Implantation

2–3 months olds of female Sprague Dawley (SD) rats were used in current study to investigate biocompatibility after implanting various nano-materials following the guidelines which were approved by Animal Care and Use Committee (La-1071565). The weight of rats was approximately 300–350 g for following experiments. After local anesthesia, the dorsal skin of each rat was incised for 10 mm to implant different nanomaterials. The wound tissue was resected for various investigations after 1 month. The capsule formation was examined through hematoxylin and eosin staining (H&E staining) for six sites, then the average thickness of encapsulated fibrotic tissues was semi-quantified by using Image J 5.0 software. A Masson’s trichrome staining kit (Sigma) was applied to investigate collagen deposition, and the fibrosis tissue area incised from each rat was analyzed through Image J 5.0 software. To further evaluate macrophage polarization, primary anti-CD86 and anti-CD163 antibody (1:200 dilution) (Santa Cruz) were applied to evaluate the M1/M2 polarization. Further, the endothelialization marker CD31 was used to examine the differentiation capacity. Anti-Mouse Alexa Fluor^®^ 488 secondary antibodies were applied for signal amplification. DAPI was applied to locate cell nuclei. A fluorescence microscope was used to record fluorescent intensity and semi-quantitative results were measured by Image J 5.0 Pro Plus (Media Gybertics). Data were represented as mean ± *SD* (*n* = 6).

### 4.12. Statistical Analysis

The experiments in current research were independently performed in triplicate to avoid uncertainty. Data for each test (*n* = 3–6) were presented as mean ± standard deviation (SD). The single-factor Analysis of Variance (ANOVA) and student *t*-test methods were applied to reveal differences among various materials. Bonferroni was used for post hoc test for ANOVA. A *p* value less than 0.05 was considered statistically significant.

## 5. Conclusions

FN-AgNP nanocomposites containing various concentrations of AgNP (~15.1, ~30.2 and ~75.5 ppm) were created in an attempt to investigate the biocompatibility and biological performances of mesenchymal stem cells. Our findings indicated that FN-AgNP composites with 30.2 ppm of AgNP enhanced the cell viability of MSCs, and inhibited monocytes activation as well as attenuate ROS generation. Besides, FN-AgNP 30.2 ppm remarkably suppressed the expression of inflammatory cytokines (TNF-*α*, IL-1β, IL-6) in MSCs. The adhesion, migration, and differentiation ability of MSC on FN-AgNP 30.2 ppm were significantly greater. The MMP enzymatic activities and expression of SDF-1α protein were also examined. The highest expression levels of MMP9 and SDF-1α were found from MSC on FN-AgNP 30.2 ppm. In vivo implantation showed that FN-AgNP 30.2 ppm had the lowest capsule formation, collagen deposition, and immune response. Moreover, the endothelial differentiation ability was enhanced by FN-AgNP nanocomposites in vivo. FN-AgNP nanocomposites may provide effective surface modification applications for vascular tissue regeneration.

## Figures and Tables

**Figure 1 ijms-22-09262-f001:**
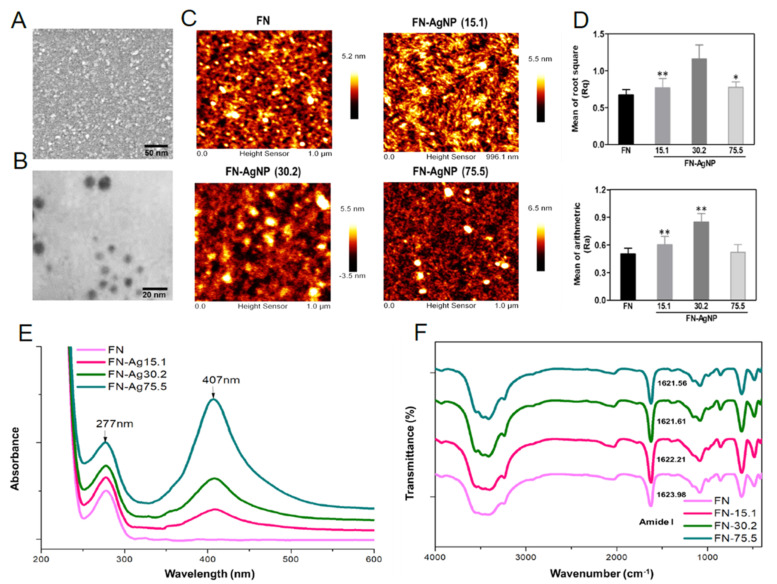
Materials characterization. The preparation procedure of fibronectin-silver nanoparticle (FN-Ag) composites. (**A**) SEM images of AgNP and (**B**) TEM image of the diameter of AgNP was about 5 nm. (**C**) Preparation of FN-AgNP nanocomposites. The surface topography of pure FN and FN-AgNP containing various amount of Ag were observed by AFM. (**D**) Rq is the roughness, while Ra represents as the average roughness of each material. * *p* < 0.05, ** *p* < 0.01: greater than the control (FN). (**E**) UV-Visible absorption peak of pure FN and FN-Ag containing various amount of AgNP (~15.1 ppm, ~30.2 ppm and ~75.5 ppm). (**F**) FTIR spectrum of pure FN and FN-Ag nanocomposites in the total wave number 400 cm^−1^ to 4000 cm^−1^. Data results from one representative experiment of three independent experiments.

**Figure 2 ijms-22-09262-f002:**
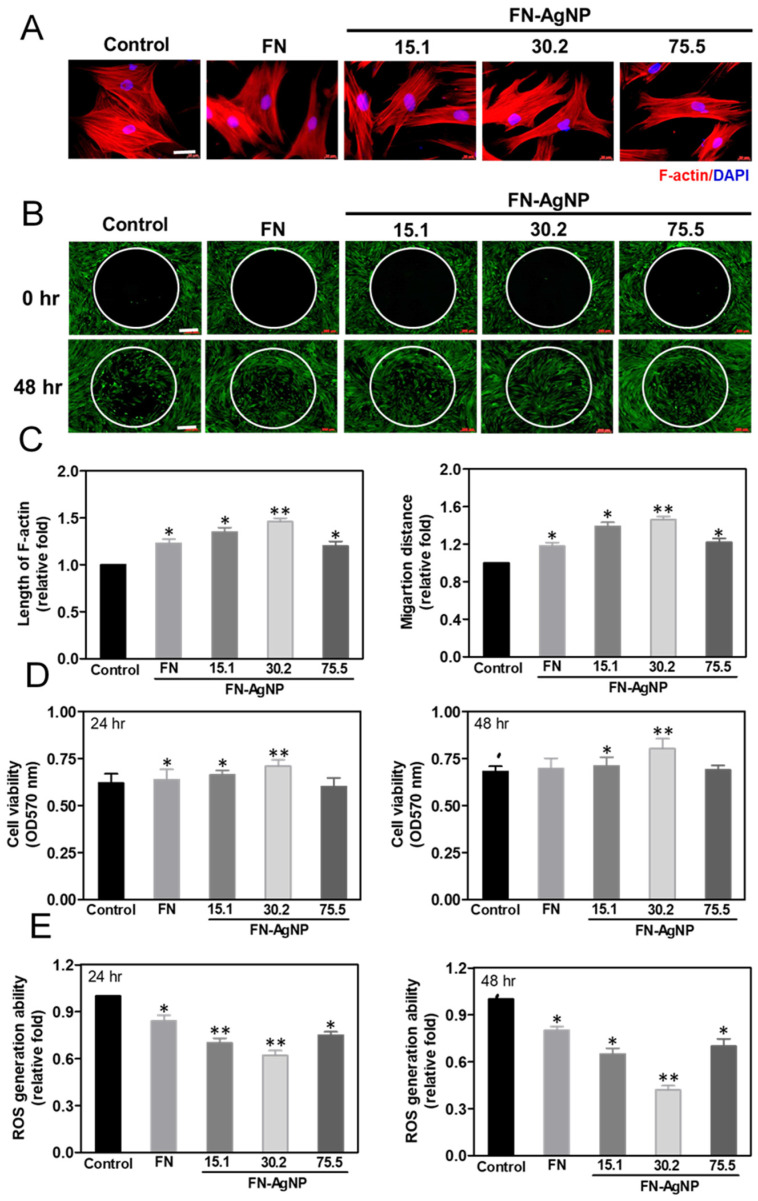
Cytoskeletal fibers examination and Migration ability of MSCs. (**A**) The cytoskeleton stain-ing for the actin fibers of MSCs cultured with pure FN and various concentrations of FN-AgNP nanomaterials at 48 h observed by fluorescence microscopy. F-actin: red color fluorescence, DAPI: blue color fluorescence. Scale bar equals to 50 μm. (**B**) MSCs migrated into central area was confirmed through fluorescence microscopy. After incubating for 24 and 48 h, calcein-AM (2 μM) solu-tion was used to stain the MSCs prior to observation. Scale bar = 200 μm. (**C**) The length of actin fiber was measured for MSCs on various nanomaterials and the results were semi-quantified at 48 h. The migration distance of MSCs seeding on various materials was semi-quantified at 48 h. Data are mean ± SD (*n* = 3). * *p* < 0.05; ** *p* < 0.01: greater than the control. (**D**) MSC proliferation after seeding on pure FN and FN-AgNP nanocomposites containing ~15.1 ppm, ~30.2 ppm and ~75.5 ppm of AgNP was investigated by MTT assay. Data are the mean ± SD (*n* = 6). * *p* < 0.05; ** *p* < 0.01: greater than the control (MSC). Data results from one representative experiment of six independent experiments The ROS production was targeted by DCFH-dA and was quantified by FACS analysis for (**E**) MSC on various materials. Data are mean ± SD (*n* = 3). * *p* < 0.05; ** *p* < 0.01: smaller than the control. Data results from one representative experiment of three independent experiments. Control group = tissue culture plate (TCPS).

**Figure 3 ijms-22-09262-f003:**
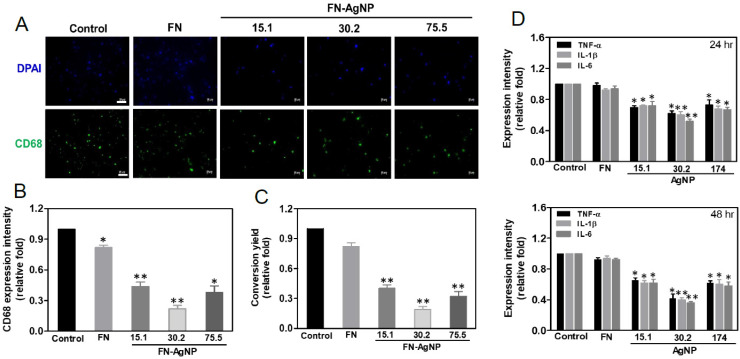
CD68 expression of macrophages on various materials at 96 h and the expression of pro-inflammatory cytokines (TNF-α, IL-1β, IL-6) at 24 and 48 h. (**A**) The cells were stained by primary anti-CD68 antibody and conjugated with FITC-immunoglobin secondary antibody (green color). DAPI solution was used to locate cell nuclei (blue color). The results were captured using fluorescence microscopy. Scale bar = 20 μm. (**B**) The expression level of CD68 was analyzed based on fluorescence intensity. (**C**) The conversion yield of human monocytes activation after culturing on pure FN and various concentrations of FN-AgNP for 96 h. (**D**) The expression of proinflammatory cytokines at 24 and 48 h was analyzed based on the expression intensity. Data are the mean ± SD (*n* = 3). * *p* < 0.05; ** *p* < 0.01: smaller than the control. Control group = glass cover slide.

**Figure 4 ijms-22-09262-f004:**
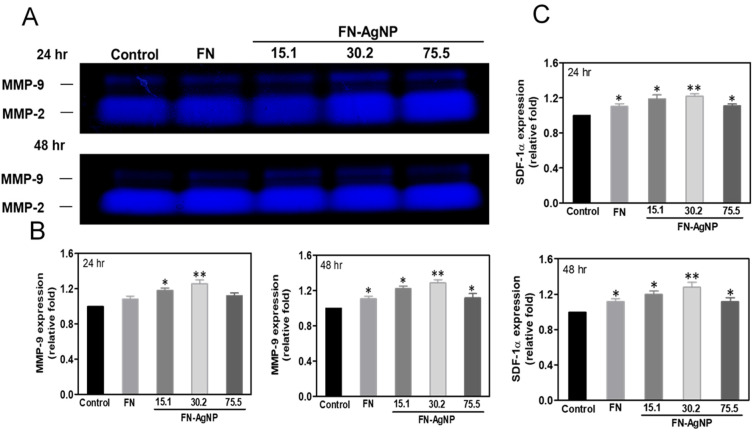
The MMP enzymatic activities and SDF-1α protein expression in MSCs culturing on various materials at 24 and 48 h. (**A**) The representative MMP-2 and MMP-9 zymogram for (a) 24 and (b) 48 h is shown. The semiquantitative meas-urement of (**B**) MMP-9 and (**C**) SDF-1α protein expression revealed a significantly greater expression culturing MSCs with FN-AgNP 30.2 ppm at 24 and 48 h. Data are presented as mean ± SD (*n* = 3). * *p* < 0.05; ** *p* < 0.01: greater than the control. Control group = glass cover slide.

**Figure 5 ijms-22-09262-f005:**
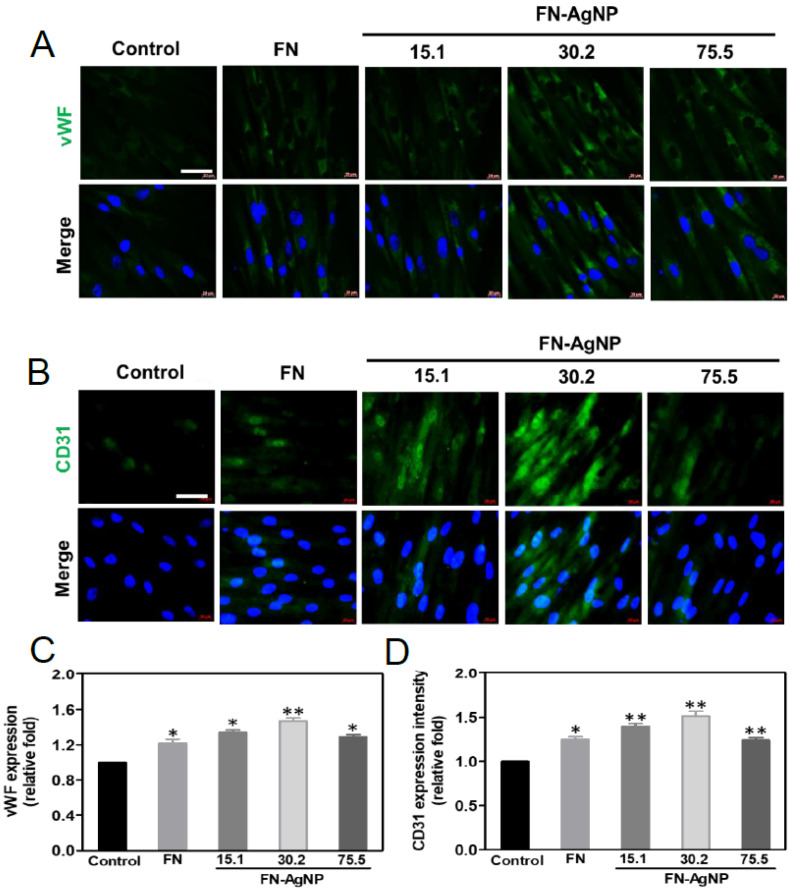
Differential expression of CD31 and vWF protein in MSCs culturing with various nanomaterials at day seven. The MSCs were firstly stained by primary (**A**) anti-CD31 and (**B**) anti-vWF antibody and conjugated with FITC secondary antibody (green color). The cell nucleus was located by DAPI (blue color). The images were taken by fluorescence micros-copy. Scale bar = 20 μm. (**C**,**D**) The quantitative results of fluorescence intensity for CD31 and vWF revealed a higher expression amount in FN-AgNP 30.2 ppm group. Data are the mean ± SD (*n* = 3). * *p* < 0.05; ** *p* < 0.01: greater than the control. Control group = TCPS.

**Figure 6 ijms-22-09262-f006:**
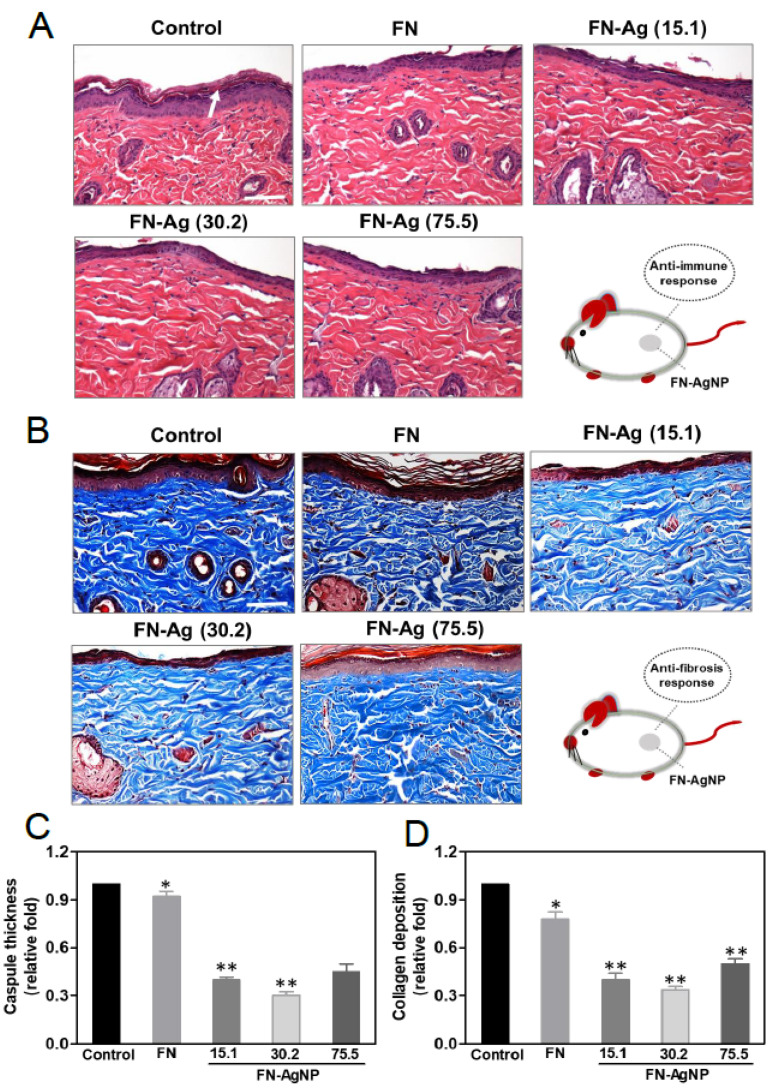
Subcutaneous implantation to measure the foreign body reactions induced by various nanomaterials. After four weeks of implantation, (**A**) H&E staining, white arrows indicate the thickness of the capsule, and (**B**) Masson’s staining were applied to process further investigations. Scale bar = 100 μm. (**C**,**D**) Semi-quantification of capsule thickness and collagen deposition was revealed based on the histology examination. Control group = glass cover slide. * *p* < 0.05; ** *p* < 0.01: less than the control (*n* = 5).

**Figure 7 ijms-22-09262-f007:**
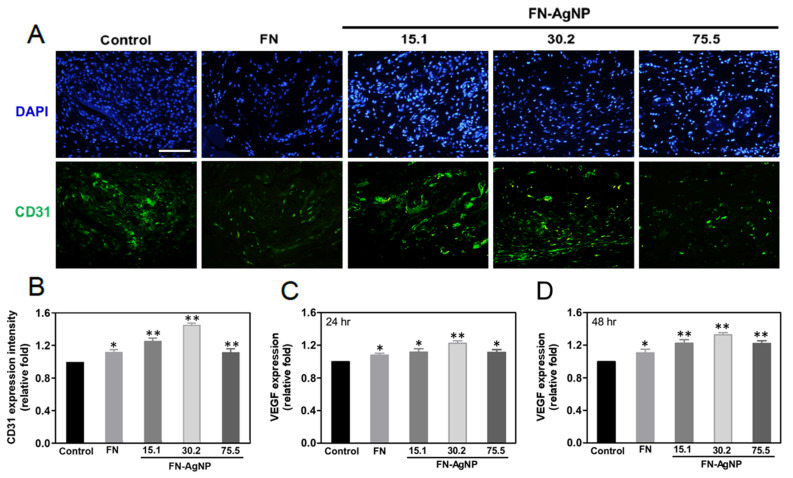
Immunohistochemistry staining for endothelialization marker CD31 affected by the implant materials. (**A**) The histology images for CD31 were showed after implanting various materials for 4 weeks. Scale bar = 100 μm. (**B**) The CD31 expression was also investigated and then semi-quantified according to fluorescence intensity. * *p* < 0.05; ** *p* < 0.01: higher than the control (*n* = 5). (**C**,**D**) The expression level of VEGF protein was evaluated via ELISA assay at 24 and 48 h, then the fluorescence intensity were semi-quantified by Image J software. Data are the mean ± SD. * *p* < 0.05; ** *p* < 0.01: greater than the control (*n* = 5).

**Figure 8 ijms-22-09262-f008:**
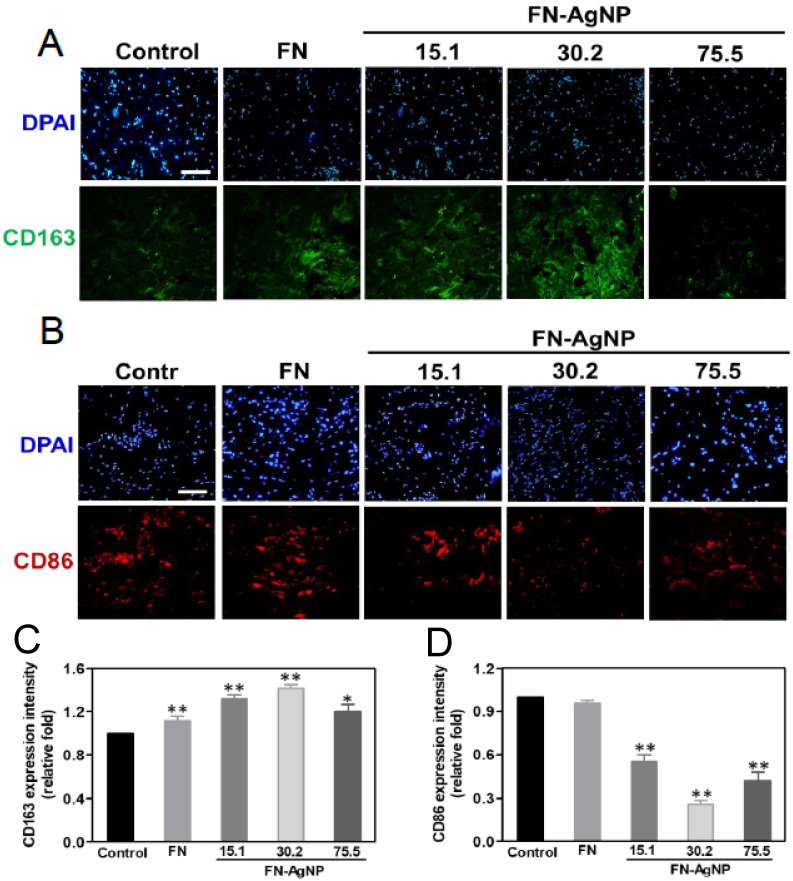
Images of immunohistochemistry staining for macrophage polarization after four weeks subcutaneous implant-ation. (**A**,**B**) The IHC staining for markers of macrophage, CD163 (M2, green color) and CD86 (M1, red color), were inves-tigated after implanting various materials. (**C**) The expression of CD163 was semi-quantified. * *p* < 0.05; ** *p* < 0.01: greater than the control (*n* = 5). (**D**) The expression of CD86 was also analyzed resulting from fluorescence intensity. * *p* < 0.05; ** *p* < 0.01: less than the control. cell nucleus was located by DAPI (blue color). Control group = glass cover slide. Scale bar = 100 μm. Data are presented as mean ± SD (*n* = 5).

**Figure 9 ijms-22-09262-f009:**
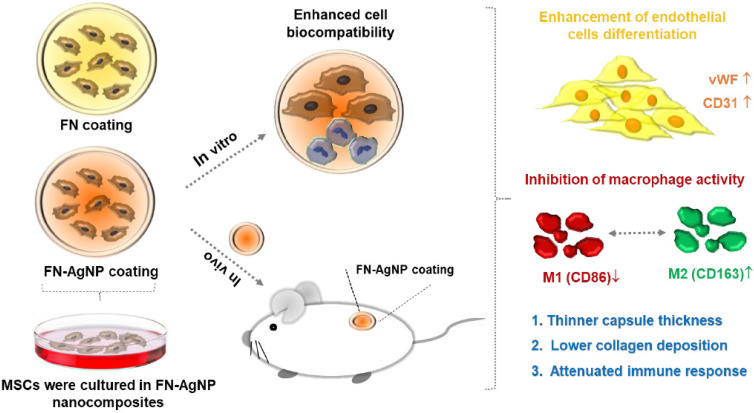
Schematic illustrations that indicate the efficacy of FN-AgNP nanocomposites. FN-AgNP nanocomposite coating enhanced the migration and differentiation ability of MSCs while inhibiting monocyte activation. Furthermore, after im-planted into the rat model, FN-AgNP induced thinner capsule formation, lower collagen deposition, and had an anti-immune response. Based on these evidences, FN-AgNP nanocomposites are suggested to be a promising surface modifi-cation strategy for biomedical applications.

## Data Availability

Not applicable.

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
