# Peer review of "Anti-Inflammatory Fibronectin-AgNP for Regulation of Biological Performance and Endothelial Differentiation Ability of Mesenchymal Stem Cells"

_ijms, 2021, doi:10.3390/ijms22179262_

Round 1

Reviewer 1 Report

In the manuscript “Anti-inflammatory Fibronectin-AgNP for Regulation of Biological Performance and Endothelial Differentiation Ability of Mesenchymal Stem Cells” authors investigated the biological compatibility of a series of Ag-fibronectin nanocomposites. They conducted the in vitro experiments using a mesenchymal cells and in vivo investigation with a rat model.  Authors demonstrate the nanocomposites facilitate the proliferation, inhibit the monocyte activation, and decreased the production of proinflammatory cytokines. Moreover, these nanocomposites promote the differentiation of endothelial cells. Manuscript is well written, but some points should be addressed by authors to improve it.

  1. The authors should explain the rationale for use 1 mg/ml FN. Was the size of nanoparticles influenced by FN? Were nanoparticles homogeneously covered by FN? May authors estimate the real FN concentrations directly associated to nanoparticles? Since the shape of AgNP is dependent of protein used (see DOI: 10.1021/acs.bioconjchem.8b00034), could authors describe with more detail this characterization?
  2. Which cellular models were used in this work? In results section, 3.2. is described the cytoskeleton changes in HSF and MSC, what is HSF? In methodology section only defined MSC and a rat model. Please clarify this point.
  3. Please, provide cytoskeleton fluorescence images with better quality. Could authors provide an example or include the description of how actin filaments were measured? How many cells were measured?
  4. To verify the accuracy of experimental methods positive controls should be included in migration assay, ROS production and cytotoxicity effect on figure 2.
  5. Authors should provide the original reference for monocytes obtention. What monocytes purity was obtained?
  6. Authors claim that Ag-FN nanocomposites reduce the monocyte activation, and they evaluate the conversion of monocyte to macrophages as a measure of monocyte activation. If CD68 was used as a marker of macrophages, why the expression was higher in the control cells? What positive control was used? Flow cytometry is a better technique to quantify surface expression molecules and characterize cellular populations, is it possible to authors provide quantitative analysis?
  7. Interesting, the Ag-FN nanocomposites induce the differentiation of mesenchymal cell to endothelial cells and this effect is more evident with FN-AgNP 30.2. Do the maximum expression of vWF and CD31 is at 7 days? Is this expression increased throughout the days?
  8. In the figure 6 authors should provide a better description of the histochemical images. What represent the white arrow in the figure 6A?
  9. Typos should be corrected. Commas are missing in some sentences.

Author Response

Comments and Suggestions for Authors

In the manuscript “Anti-inflammatory Fibronectin-AgNP for Regulation of Biological Performance and Endothelial Differentiation Ability of Mesenchymal Stem Cells” authors investigated the biological compatibility of a series of Ag-fibronectin nanocomposites. They conducted the in vitro experiments using a mesenchymal cells and in vivo investigation with a rat model.  Authors demonstrate the nanocomposites facilitate the proliferation, inhibit the monocyte activation, and decreased the production of proinflammatory cytokines. Moreover, these nanocomposites promote the differentiation of endothelial cells. Manuscript is well written, but some points should be addressed by authors to improve it.

  1. The authors should explain the rationale for use 1 mg/ml FN. Was the size of nanoparticles influenced by FN? Were nanoparticles homogeneously covered by FN? May authors estimate the real FN concentrations directly associated to nanoparticles? Since the shape of AgNP is dependent of protein used (see DOI: 10.1021/acs.bioconjchem.8b00034), could authors describe with more detail this characterization?

Answer:

(1) In a previously study, thin coating of gold nanoparticles (AuNPs) and fibronectin (FN) was developed to improve the biocompatibility required for cardiovascular devices [1].  Therefore, we followed the same concentration of FN and changed the concentrations of AgNPs in this study.  Meanwhile, the nanoparticles were generated by physical manufacturing without addition of any surface modifiers or stabilizers.  The size of nanoparticles would not be influenced by FN.

(2) In our study, FN was homogeneously mixed with the nanoparticles. The combination resulted in the absorption peak at 277nm verified by UV/visible light spectroscopy (Figure 1E). As the concentration of Ag nanoparticles increased, the absorption peak intensity also increased significantly.

(3) Thanks for your suggestion. The Ag nanoparticles used in this study are spherical particles with a diameter of 5 nm by TEM analysis (Figure 1B). On the other hand, the protein was combined after the generation of nanoparticles, so it would not affect the shape of the nanoparticles.

Reference:

  1. Hung, H.-S.; Tang, C.-M.; Lin, C.-H.; Lin, S.-Z.; Chu, M.-Y.; Sun, W.-S.; Kao, W.-C.; Hsien-Hsu, H.; Huang, C.-Y.; Hsu, S.-h. Biocompatibility and favorable response of mesenchymal stem cells on fibronectin-gold nanocomposites. PLoS One 2013, 8, e65738.

  1. Which cellular models were used in this work? In results section, 3.2. is described the cytoskeleton changes in HSF and MSC, what is HSF? In methodology section only defined MSC and a rat model. Please clarify this point.

Answer:

We have included the more detail description of HSF in the “Materials and Methods” section. “Human skin fibroblasts (HSF) were purchased from American Type Culture Collection (ATCC). They can be cultured to higher passage numbers without appreciable loss of growth rate or phenotype, thus yielding more cells for the experiments. HSF were maintained in Dulbecco’s Modified Eagle’s Medium (DMEM) (Gibco) supplemented with 1% (v/v) antibodies (10,000 U/mL penicillin G and 10 mg/mL streptomycin), 2 mM glutamine, and 10% fetal bovine serum (FBS).” (Page 4, line 161-166) 

  1. Please, provide cytoskeleton fluorescence images with better quality. Could authors provide an example or include the description of how actin filaments were measured? How many cells were measured?

Answer:

  • We have performed and included the new data in Figure 2A.

(2) We have added the following sentences in the “Materials and Methods” section “The length of F-actin fiber in MSCs was measured as the ratio of the length of the existing active fiber to its potentially possible length restricted by cell margins. Length of F-actin fiber in MSCs were quantified using Image J (https://rsb.info.nih.gov/ij) (version 1.8.0_172) software by obtaining data from five different positions for each cell in the measurement (n=3)” (Page 5, line 203-207)   

  1. To verify the accuracy of experimental methods positive controls should be included in migration assay, ROS production and cytotoxicity effect on figure 2.

Answer:

We have corrected the typo of “MSC” to “Control” (TCPS) in Figure 2.

  1. Authors should provide the original reference for monocytes obtention. What monocytes purity was obtained?

Answer:

  • We have replaced the original reference [35] and removed reference [44].

(2) We have included more detail description of monocytes into the “Materials and methods” section “the cell concentration was adjusted to 1´105 cells/ml” (Page 4, line 186)

  1. Authors claim that Ag-FN nanocomposites reduce the monocyte activation, and they evaluate the conversion of monocyte to macrophages as a measure of monocyte activation. If CD68 was used as a marker of macrophages, why the expression was higher in the control cells? What positive control was used? Flow cytometry is a better technique to quantify surface expression molecules and characterize cellular populations, is it possible to authors provide quantitative analysis?

Answer:

We thank the valuable comment from the reviewer. The control group used was the glass cover slip. The adherence to glass is commonly used to prepare monocyte monolayers from blood mononuclear cell suspensions. It was suggested that monocytes are also able to adhere to glass or glass substitutes like nylon. Monocyte monolayers prepared by adherence of blood mononuclear cells to glass or tissue culture dishes may also be contaminated by lymphocytes that subsequently induce the activation of monocytes into macrophages. Regarding this report, the glass cover slide was used as a positive control in this assay.

Reference:

Charles A. Koller, Gerald W. King, Paul E. Hurtubise, Arthur L. Sagone and Albert F. LoBuglio J Characterization of Glass Adherent Human Mononuclear Cells, Immunol, 1973, 111 (5) 1610-1612.

  1. Interesting, the Ag-FN nanocomposites induce the differentiation of mesenchymal cell to endothelial cells and this effect is more evident with FN-AgNP 30.2. Do the maximum expression of vWF and CD31 is at 7 days? Is this expression increased throughout the days.

Answer:

We have included the data in the new Figure S3 and description in the “Results” section “The differentiation capacity of MSCs induced by various materials was then evaluated. The phenotypes of endothelial cells were also observed at days 3 and 5. After culturing MSCs with various materials, FN-Ag 30.2 ppm induced a slight expression level of endothelial markers at days 3 and 5 as compared to the control, with the immunostaining images shown in Figure S3A-3B. The semi-quantitative data showed that the mineral?? differentiation in the FN-Ag 30.2 ppm group was the greatest at days 3 and 5 (p < 0.01) (Figure S3C-D).” (Page 12, line 407-414) and “Figure caption” section “Figure S3. Differential expression of vWF and CD31 and protein in MSCs culturing with various nanomaterials at days 3 and 5. The MSCs were firstly stained by primary (A) anti - CD31 and (B) anti - vWF antibody and conjugated with FITC secondary antibody (green color). The cell nucleus was located by DAPI (blue color). The images were taken by fluorescence microscopy. Scale bar = 20 μm. (C-D) The quantitative results of fluorescence intensity for vWF and CD31 revealed a higher expression amount in FN-AgNP 30.2 ppm group. Data are the mean ± SD (n = 3). *p<0.05; **p<0.01: greater than the control.” (Figure S3)

  1. In the figure 6 authors should provide a better description of the histochemical images. What represent the white arrow in the figure 6A?

Answer:

(1) We have modified the wording of Figure 6A and Figure 6B. “The fibrotic encapsulation caused by foreign body response from the different materials was observed via subcutaneous after one month of implantation in order to confirm the biocompatibility and biosafety in vivo (Figure 6A). Indeed, it was also calculated the intensity of tissue fibrosis effect using Masson’s trichrome staining, which revealed collagen deposition in response to the control treated group (glass) (Figure 6B).” (Page 14, line 426-431)

(2) We have also included the detailed description of white arrows in the “Figure caption” section “white arrows indicate the thickness of the capsule.” (Page 16, line 459)  

  1. Typos should be corrected. Commas are missing in some sentences.

Answer:

Thanks the valuable comment from the reviewer. We have proofread the article to make it well understood by the readers.

Reviewer 2 Report

Lines 287-288: when you say that " there was a shift in the peak position of amide I from 1623.98 cm-1 (pure FN) 287 to 1622.21 cm-1 (FN-AgNP 15.1 ppm), 1621.61 cm-1 (FN-AgNP 30.2 ppm) and 1621.56 cm-1 288
(FN-AgNP 75.5 ppm)"...; Can you confirm these data with literaure? 

Lines 506-508 Correct the font size

Line 515-516: you assert that "the lowest fibrous encapsulation and collagen deposition, indicating the lowest biotoxicity of FN-AgNP 30.2"...; Why don't justify this statement with an in vitro test of cellular toxicity?

Author Response

Comments and Suggestions for Authors

  1. Lines 287-288: when you say that " there was a shift in the peak position of amide I from 1623.98 cm-1 (pure FN) 287 to 1622.21 cm-1 (FN-AgNP 15.1 ppm), 1621.61 cm-1 (FN-AgNP 30.2 ppm) and 1621.56 cm-1 288 (FN-AgNP 75.5 ppm)"...; Can you confirm these data with literature ?

Answer:

Thanks for your comment. The above sentences were modified to, “According to Figure 1F,  the position of the amide-I band maximum was at approximately 1636 cm-1 [44]. When FN was combined with AgNP, there was a shift in the peak position of amide I from 1623.98 cm-1 (pure FN) to 1622.21 cm-1 (FN-AgNP 15.1 ppm), 1621.61 cm-1 (FN-AgNP 30.2 ppm) and 1621.56 cm-1 (FN-AgNP 75.5 ppm). The above findings indicated that the amide I band may have strong interaction with AgNP.” (Page 6, line 293-297)

  1. Lines 506-508 Correct the font size

Answer:

We have corrected the font size of “polyurethane-AuNPs (PU-AuNPs) could regulate ECs migration via triggering α5β3 integrin/FAK and PI3K/Akt/eNOS signaling pathways”. (Page ?, line ?)

  1. Line 515-516: you assert that "the lowest fibrous encapsulation and collagen deposition, indicating the lowest biotoxicity of FN-AgNP 30.2"...; Why don't justify this statement with an in vitro test of cellular toxicity?

Answer:

  • We have performed the cell viability test and presented the data in Figure 2D.

(2) We have changed the wording “indicating the lowest biotoxicity of FN-AgNP 30.2” to “indicating the better biocompatibility of FN-Ag 30.2” (Page 19, line 532-533)

Round 2

Reviewer 1 Report

The authors of the manuscript “Anti-inflammatory Fibronectin-AgNP for Regulation of Biological Performance and Endothelial Differentiation Ability of Mesenchymal Stem Cells” have positive replied to my main criticism; however, some points still need to be clarified.  

  1. Authors should explain what’s the reason for use human skin fibroblasts and should discuss the results obtained from these cells.
  2. Authors should clarify what conditions were used as controls in all figure legends.

Author Response

1. Authors should explain what’s the reason for use human skin fibroblasts and should discuss the results obtained from these cells.

Answer:

We thank the valuable comment from the reviewer. We have included more detail description in the “Results” section “It has been reported that co-culture of fibroblasts and endothelial could significantly enhance the angiogenesis of endothelial cells. It was indicated that there are intimate communications between fibroblasts and endothelial cells for angiogenesis, and the paracrine effect may play a crucial role in these communications [46]. Therefore, the effects of different materials on biocompatibility and the biological activity of fibroblast (HSF) was also investigated in this study.” (Page 7, lines 310-314) and modified the “Reference” section (marked with red color)

2. Authors should clarify what conditions were used as controls in all figure legends.

Answer:

We have included the control group in each “Figure caption” section (marked with red color).

3. We also replace the new Figure 4 make the label of control group to be consistent.
